# Genetic Structure and Colonization of North America by *Depressaria depressana* (Fabricius 1775) (Lepidoptera: Depressariidae) over 15 Years; Contrasts with Westward Expansion of *Depressaria radiella* (Goeze, 1783) over 160 Years

**DOI:** 10.3390/insects13090789

**Published:** 2022-08-31

**Authors:** Charles A. E. Dean, Jack Easley, Aron D. Katz, Stewart H. Berlocher, May R. Berenbaum

**Affiliations:** 1Department of Entomology, University of Illinois Urbana-Champaign, Urbana, IL 61801, USA or; 2Engineer Research and Development Center, Champaign, IL 61821, USA; 3Illinois Natural History Survey, Prairie Research Institute, University of Illinois Urbana-Champaign, Champaign, IL 61820, USA

**Keywords:** invasion, herbivory, Apiaceae, Depressariidae, museum records

## Abstract

**Simple Summary:**

As invasive insect species can disrupt agriculture, ecosystem functions, and human health, monitoring the spread of newly invasive insects is a high priority. The purple carrot-seed moth *Depressaria depressana*, a European species first reported in North America in 2008, specializes on reproductive structures of taxa throughout the family Apiaceae and thus presents a potential threat to native North American umbellifers. We assessed interpopulational genetic diversity of *D. depressana* across its eastern North American invasion front and compared those values to available data for European populations. We also compared this diversity with genetic diversity estimates for populations of *Depressaria radiella* (parsnip webworm), a European species introduced to North America more than 160 years ago that is essentially restricted to two apiaceous genera throughout its native and invasive ranges. Documenting the historical spread of *D. radiella* with museum and literature records, we found *D. depressana* displays greater genetic diversity than *D. radiella* and is colonizing North America more rapidly; greater genetic diversity may facilitate faster colonization via establishment on a wider range of hostplants. The contrast in colonization by these two species may inform management practices for invasive insects utilizing a broad versus narrow array of hosts within a single plant family.

**Abstract:**

*Depressaria depressana*, the purple carrot seed moth, is a Eurasian species first reported in North America in 2008 and currently undergoing range expansion. This invasion follows that of its Eurasion congener *Depressaria radiella* (parsnip webworm), first documented in North America 160 years ago. Unlike *D. depressana*, which utilizes hostplants across multiple tribes of Apiaceae, *Depressaria radiella* is a “superspecialist” effectively restricted in its native and non-indigenous ranges to two closely related apiaceous genera. We investigated the genetic structure of *D. depressana* populations across latitudinal and longitudinal gradients in the eastern United States by constructing COI haplotype networks and then comparing these with haplotype networks constructed from available COI sequence data from contemporary European *D. depressana* populations and from European and North American *D. radiella* populations. Haplotype data revealed higher genetic diversity in *D. depressana*, indicating high dispersal capacity, multiple introductions, and/or a genetically diverse founding population. Museum and literature records of *D. radiella* date back to 1862 and indicate that range expansion to the West Coast required more than 50 years. Higher levels of genetic diversity observed in *D. depressana* compared to its congener may indicate a greater propensity for dispersal, colonization and establishment in its non-indigenous range.

## 1. Introduction

A substantial increase in the number of invasive species in North America has occurred over the past 50 years as a direct result of increased globalization of trade and travel [1,2,3]. Managing the introduction of exotic insect species into new habitats is particularly problematic owing to their typical ease of transport, adaptability to new environments, rapid generation times, and high reproductive capacity [4,5]. In North America alone, there are more than 470 introduced insect species (Center for Invasive Species and Ecosystem Health, invasive.org), which annually incur costs exceeding $23 billion in the region [6]. This damage includes substantial loss of native biodiversity, loss of ecosystem services, crop yield loss, destruction of infrastructure, and declines in human health. Developing successful prevention and management practices is predicated upon a thorough understanding of both the ecological and evolutionary processes underlying insect colonization of novel habitats. To that end, monitoring the spread of a recently introduced species offers a unique opportunity to develop models of insect dispersal across, and adaptation to new environments.

The purple carrot-seed moth *Depressaria depressana* (Fabricius 1775) (Lepidoptera: Depressariidae), native to Eurasia, was first documented in the scientific literature as an introduced species in the United States in 2015 [7] and has likely been established in North America since at least 2008 [8], with images of larvae and adults appearing on BugGuide from New York, Connecticut, and Massachusetts as early as 2010. Although restricted to hostplants in the family Apiaceae, *D. depressana* utilizes species across a wide taxonomic cross-section of the family [7]. Moreover, in contrast with most *Depressaria* species, it is multivoltine and exhibits seasonal activity from late spring to early fall in central Illinois (C. Dean and M. Berenbaum, personal observations).

In expanding its range to include North America, *D. depressana* follows its Eurasian congener *Depressaria radiella* (Goeze, 1783) (formerly *Depressaria pastinacella* Duponchel, 1838), which was first reported in the Western Hemisphere in Ontario in 1862 (Bethune 1869, named *D. ontariella* and subsequently synonymized with the European *D. heracliana* (Linnaeus, 1758)). Like *D. depressana*, *D. radiella* is associated with hostplants in the family Apiaceae, but it is restricted to the two closely related genera, *Heracleum* Linnaeus, 1753 and *Pastinaca* (Linnaeus 1753), in North America and is univoltine throughout its range. Early reports of its occurrence in North America were associated with the damage it inflicted on flowers and fruits of *Pastinaca sativa* (Linnaeus 1753), the edible parsnip [9,10,11,12,13]. *P. sativa* is an apiaceous weedy biennial native to Eurasia thought to have been domesticated as a root vegetable centuries ago. Both wild and domesticated forms were introduced to North America in the seventeenth century [14] and the range of the interaction between parsnip webworms and the wild parsnip now extends across much of Canada and the United States [14].

Examination of *P. sativa* sampled from herbarium collections revealed that North American wild parsnip populations increased in toxic furanocoumarin content shortly after the accidental introduction of *D. radiella* to North America [15] and that, within its midwestern US range, parsnip webworms exhibit population-level detoxification profiles reflecting the furanocoumarin profiles in the immature fruits of their corresponding hostplant populations [16]. The reestablishment of the *D. radiella* and *P. sativa* system across novel landscapes has since been repeated in New Zealand. In 2004, *D. radiella* was first documented in Dunedin roughly 150 years after the introduction of *P. sativa* [17]. Studies conducted over a six-year period after the introduction of *D. radiella* to New Zealand found that *P. sativa* fitness was adversely affected following a reduction in seed production by up to 75% in six previously uninfested populations. As a compensatory mechanism, *P. sativa* populations in New Zealand, which were distinct from European and North American populations in phytochemical profile, exhibited increased growth rate accompanied by greater production of the floral volatile octyl butyrate [18].

For more than 40 years, the relationship between wild parsnip and the parsnip webworm *D. radiella* has served as a useful model for studying exotic plant-insect reassociations within and across novel landscapes [19]. While the colonization of *D. radiella* across multiple continents provides an exemplar of rapid ecological and evolutionary change following establishment of an invasive species, it is inherently constrained by the tightly coevolved nature of the *D. radiella-P. sativa* relationship. Since its introduction approximately 160 years ago, for example, *D. radiella* has acquired only a single native hostplant, *Heracleum maximum*, American cow-parsnip, the sole North American congener of many of its European hostplants [20]. *D. depressana*, however, utilizes a broader array of host plants in its native range species and has been associated with at least a dozen additional apiaceous genera in a diversity of tribes across Eurasia (Dean et al. submitted). Moreover, the extended period of seasonal activity of *D. depressana*, combined with its status as a family level specialist, suggests that, while the selection pressure exerted on its many hostplants may be less intense than that exerted by its congener, the broader host range and extended period of seasonal activity suggest that this species has a greater likelihood of encountering and acquiring native North American plants as novel hosts (Dean et al. submitted). Investigating the colonization of the family specialist *D. depressana* across North America can serve as a unique comparative measure against that of the “superspecialist” *D. radiella* (restricted to only two genera) and may inform predictions of the likely establishment and potential selective impact of invasive herbivores as a function of diet breadth.

A critical component of tracking ongoing biological invasions is the assessment of populational genetic structure along putative routes of establishment, which can facilitate inferences as to the source population, mode of introduction, and rate of evolution. In this study, we used the molecular marker mitochondrial cytochrome c oxidase subunit I (COI) to assess the population genetic structure of *D. depressana* in the eastern United States. Using this marker, we constructed haplotype networks to evaluate the inter- and intra-populational genetic diversity across both latitudinal and longitudinal gradients to the north and east of Illinois, where *D. depressana* was first reported in the United States [7]. We also constructed a haplotype network incorporating both European and North American *D. radiella*, using COI barcodes available in the BOLD [21] database. Lastly, using historical publications and museum records, we reconstructed an approximate timeline of the range expansion of *D. radiella* across North America. Together, the COI haplotype network and the approximate invasion timeline of the superspecialist *D. radiella* can provide a comparative insight as to whether the ecological differences in diet breadth and period of seasonal activity influence population structure across geographic space and whether these impacts are likely to influence the rate of successful *D. depressana* establishment across North America.

## 2. Materials and Methods

### 2.1. D. depressana Sampling

*D. depressana* larvae were sampled from the umbels of wild carrot *Daucus carota* at eight different collection sites along a latitudinal gradient from Urbana, IL to Eau Claire, WI in July 2020 and nine collection sites along a longitudinal axis from Urbana, IL to Newburg, PA in July 2021. Sample sites were selected by distance and collections occurred every 1 h of driving, or approximately every 70 miles (113 km). Ten ultimate instar larvae were collected from each site and preserved in labeled glass tubes filled with 90% EtOH. The state and GPS coordinates corresponding to each collection site are given in Table 1. These GPS locations are presented in maps in Figure 1.

### 2.2. DNA Extractions and PCR Amplification

For freshly collected specimens, total genomic DNA was collected from whole-body tissue extractions of *D. depressana* using the Qiagen DNeasy Blood and Tissue Kit (Qiagen Inc., Valencia, CA, USA) following the manufacturer’s protocol. Ultimate instar caterpillars were individually homogenized in a solution containing buffer ATL and proteinase K using an autoclavable pestle (Fisher Scientific, Hampton, NH, USA). The homogenized tissue was incubated at 56 °C for 1 h to ensure complete cell lysis. A series of wash buffers was used to remove RNA, proteins, and other contaminants from the solution before DNA was ultimately eluted in 50 µL of ddH_2_O. DNA concentration was assessed via Nanodrop spectrophotometer (Thermo Fisher Scientific, Waltham, MA, USA).

PCR amplification was performed for mitochondrial COI. The partial sequence of COI was amplified with the following primers from Folmer et al. (1994) [22]: (LCO1490) 5′ GGTCAACAAATCATAAAGATATTGG 3′ and (HCO2198) 5′ TAAACTTCAGGGTGACCAAAAAATCA 3′ generating a 658 bp amplicon. Onetaq polymerase was used for PCR reactions (New England Biolabs, Ipswich, MA, USA). PCR conditions were 95 °C/1 min for the initial denaturation phase, then 40 cycles of 95 °C/30 s, 50 °C/30 s, 68 °C/1 min, followed by a final extension phase at 72 °C/5 min. Sanger sequencing was performed at the Keck Center at University of Illinois, Urbana-Champaign.

Additionally, COI barcodes were sourced from the BOLD database [21] to be included in the COI haplotype network as a comparative measure to evaluate potential source populations. Similarly, COI barcodes for *D. radiella* were also obtained from BOLD for reconstruction of European and North American haplotypes (http://v4.boldsystems.org/index.php/Public_BINSearch?searchtype=records, accessed on 25 June 2022). BOLD IDs and Genbank accession numbers for all COI barcodes generated and retrieved as part of this analysis are given in Appendix A.

### 2.3. Alignment and Gene Characterization

Raw sequence reads were imported to Geneious v.2021.1.1 (Dotmatics, Boston, MA, USA) and ends were trimmed using an error probability limit of 0.05. Complementary paired sequences were assembled de novo, read quality was visually assessed by distinctness of fluorescence peaks. The consensus regions were then extracted, and primer regions were trimmed. All COI sequences were aligned in Geneious v.2021.1.1 (Dotmatics, Boston, MA, USA) via MAFFT v 7.450 [23] using default parameters.

### 2.4. Haplotype Network Construction and Analysis

Partial *D. depressana* and *D. radiella* sequences for COI were exported separately as Nexus files that included trait blocks incorporating sampling site assignments. Haplotype designations and networks were generated in PopART v.1.7 [24] using the Templeton-Crandall-Sing (TCS) algorithm [25]. Genetic structuring across sampling sites was assessed across geographic space via analysis of molecular variance (AMOVA) with PopART v.1.7 [24]. A Mantel test was also performed using zt v.1.1 [26] to evaluate the relationship between genetic variation and geographic distance.

Publications documenting the earliest observations of *D. radiella* in North American states and provinces were retrieved from PubMed.gov. Search terms “*Depressaria pastinacella*” “*Depressaria heracliana*” and “*Depressaria ontariella*” were used and the results were filtered by date. Additionally, physical copies of publications documenting early observations of *D. radiella* in North America were included in this analysis. These publications were retrieved from the private collection of one of the authors (MRB). Searches for *D. radiella* museum records were performed in iDigBio.org using the query terms “*Depressaria radiella*” or “*Depressaria pastinacella*” with United States and Canada as the “Country” specification. Search results were filtered by “Date” and “State/Province” of specimen collection. The earliest museum record of every state or province was used in the analysis and the corresponding museum was noted.

## 3. Results

### 3.1. Depressaria Depressana Haplotypes

The COI locus for 167 *D. depressana* individuals was amplified and sequenced. Additionally, 34 sequences of COI were retrieved from the BOLD database and added to the alignment containing COI sequences generated in-house for this analysis. Across the ~680-km latitudinal gradient ranging from Urbana, IL, to Eau Claire, WI, 79 COI sequences were generated from *D. depressana* samples collected across eight geographic locations. COI sequences from the 34 additional taxa retrieved from the BOLD database were incorporated into the 658-bp nucleotide alignment. Uncorrected pairwise COI distances revealed genetic distances of 0–2%. Within this alignment, 22 polymorphic sites were identified as informative for the delineation of haplotypes, and individuals were subsequently clustered into eight distinct haplotypes (Figure 2). A low level of genetic diversity was detected across all samples and the AMOVA revealed little genetic structuring (ɸ_ST_ = 0.095, *p* = 0.059).

Analysis of the ~1078 km longitudinal gradient ranging from Urbana, IL, to Newburg, PA, incorporated 88 *D. depressana* samples collected from nine geographic locations. For these specimens, 88 COI sequences were generated and aligned to the 34 COI sequences retrieved from the BOLD database. Seven informative polymorphic sites were identified, and taxa were clustered into six distinct haplotypes. Similarly, a low level of genetic structure was observed (AMOVA, ɸ_ST_ = 0.12, *p* = 0.068) (Figure 3).

Combining the *D. depressana* COI datasets obtained from samples collected along the latitudinal and longitudinal gradients revealed 23 informative polymorphic sites, clustering the taxa into 5 haplotypes with 4 shared between gradients. An AMOVA run on the pooled dataset also revealed a low level of genetic structure (ɸ_ST_ = 0.10, *p* = 0.058) (Figure 4). The Mantel test returned a significant but very weak positive correlation between geographic distance and genetic variation (r = 0.049, *p* = 0.042), indicating that isolation by distance has little to no effect of genetic variation.

### 3.2. Depressaria radiella Haplotypes

With respect to variation at the COI barcode fragment in European and North American *Depressaria radiella,* analysis of the 658-bp region of COI for *D. radiella* incorporated 45 individuals. Twenty-six complete COI sequences were found, 16 from Europe and 10 from North America. In addition, 19 partial sequences were found, 6 from Europe and 13 from North America. The 26 complete DNA barcodes were aligned with NCBI Multiple Sequence Alignment ViewEr 1.22.0. It is clear from Table 2, the resulting haplotype network revealed two distinct haplotypes, with 5 informative SNPs distinguishing them (Table 2, Figure 5). All individuals fell into the H1 haplotype, with the single *D. radiella* specimen collected in Russia constituting the H2 haplotype.

### 3.3. Reconstruction of the History of Invasion of North America by D. radiella

The literature search results and museum records used in establishing the timeline of the westward expansion of *D. radiella* across North America are presented in Table 3. The earliest recording of *D. radiella* in North America occurred in Ontario in 1862 and the earliest recording of the presence of *D. radiella* in the United States occurred in 1888 in Bristol, Pennsylvania, 26 years later. Records show that, by 1889, *D. radiella* was successfully established across most of the northeastern United States. *D. radiella* was first observed on the west coast of the continent in Oregon in 1914, approximately 50 years after its presence in North America was first recorded.

## 4. Discussion

COI is a widely used marker for assessing haplotype diversity in invasive and migratory Lepidoptera [27,28,29,30,31]. This gene evolves at a relatively slow rate in the class Insecta [32]. COI sequencing offers a reliable method of evaluating competing hypotheses regarding invasive populations and their introduction pathways into new geographic areas. Sampling both field-collected and museum specimens of *Helicoverpa armigera* across Argentina, Balbi et al. (2020) [27] uncovered 5 COI haplotypes, along with 3 cytochrome b (Cytb) haplotypes, which provided preliminary evidence of multiple points of introduction into the region from neighboring Latin American countries. Similarly, Lee et al. (2020) [31] proposed that, among populations of *Spodoptera frugiperda* (J.E. Smith), a species recently introduced to South Korea, the two distinct COI haplotypes present were suggestive of Brazil populations as the source of this invasion. This study was the first to monitor the genetic distribution of *S.*
*frugiperda*, which, given its high capacity for migration and dispersal, makes it a candidate to become a major agricultural pest in South Korea [26]. In our study, we found 8 COI haplotypes among the eight sites spanning the latitudinal gradient and six haplotypes among the nine sites spanning the longitudinal gradient with 4 shared between gradients. Across all COI sequences, we found that populations of *D. depressana* established within the United States maintain relatively high levels of genetic diversity compared to *D. radiella*. The lack of strong genetic structure observed among sampling sites could also be an indication that the entirety of the sampling area is effectively one population (i.e., *D. depressana* is experiencing high gene-flow or rapid expansion). Although a larger sampling area might have been needed to detect genetic structure, these data clearly indicate that *D. depressana* are excellent dispersers. Multiple recent introductions throughout the country combined with high dispersal capacity would effectively obscure any genetic structure among sampling sites. The Mantel test further supports this assertion by providing significant evidence that geographic distance is has no major effect on patterns genetic variation–an indication that their propensity to disperse long distances is not driving genetic isolation.

Our findings are consistent with either multiple introductions of *D. depressana* or a single introduction of a large population retaining a large amount of diversity of ancestrally polymorphic alleles. While the dataset sampled from the BOLD database was limited, 6 haplotypes were recovered from the 35 sequences originating in Europe and Canada. However, the populations sampled in the United States do not strictly coincide to specific European or Canadian populations, consistent with ongoing interpopulational outcrossing, high levels of gene flow, or a lack of sufficient evolutionary time for geographic sorting of COI haplotypes. Multiple introductions may indeed provide an adaptive advantage to *D. depressana*, as it reduces the likelihood of bottleneck events.

In contrast, the COI haplotype network for *D. radiella* reveals relatively little genetic diversity within North America and across Europe, with significant geographic structuring between Russian and European/North American haplotypes. All individuals included in this analysis comprise a single haplotype except for one haplotype collected in Russia. This Russian individual was placed into a separate haplotype on the basis of four informative SNPs.

This apparent lack of variation, if validated by more extensive sampling, has significant implications for understanding the biology of *D. radiella*. Lack of variation in just the introduced North American populations could be explained by a simple, rapid loss of variation in a colonization bottleneck, but the low level of variation in the European source population would require a more complex evolutionary explanation, conceivably related to the moth’s limited host range or dispersal capacity; the only information about dispersal capacity is reported by Zangerl and Berenbaum (2003) [33], who determined experimentally that gene flow via adult migration extends at least 1.3 km. However, before investing effort in such an evolutionary explanation, it is worth considering the Russian population in more detail. Several features stand out. First, the reported site, at 2600 m on Mt. Terskol in the Russian Caucasus mountains, is approximately 2000 km east of the rest of the European samples. Moreover, the wingspan of the specimen is 32 mm, well outside the reported wingspan range of *D. radiella* of 19–27 mm. Given the 5 unique bases found in this atypical specimen, the possibility that it is a new species-not *D. radiella*-should be given serious consideration.

If the Russian sample is removed, then all complete *D. radiella* sequences are identical. While entering a set of identical sequences into, e.g., PopArt returns a ɸFST value of “nil”, for discussion such a ɸFST can be considered as 0 between the source (Europe) and colonized (North American) populations. Given this situation, the partial sequences might provide insights. For one thing, two additional haplotypes are revealed. The Croatian specimen and one of the Ontario, Canada specimens each has a unique haplotype (although these two haplotypes share the base change at position 628). These haplotypes indicate that further sequencing of *D. radiella* would reveal additional variation, although the available data suggest that such an increase in variation would be modest. Finally, the fact that one of the additional haplotypes is from Europe and the other is from North American suggests that more sampling could continue to show roughly equal amounts of variation in source and colonist samples.

Nevertheless, the apparent low genetic diversity may indicate a slower rate of molecular evolution in *D. radiella* compared to *D. depressana*. Multiple generations per year and a lengthier period of seasonal activity in *D. depressana* may be predicated upon its expanded host range, which comprises apiaceous hosts that have sequential yet overlapping blooming periods from May through September. It is possible that the larger number of hosts of *D. depressana* may have selected for greater genetic variation relative to *D. radiella*; evidence for the “niche width-variation hypothesis” is mixed, with some authors reporting increased variation in more polyphagous species [34,35] and others reporting the exact opposite relationship [36,37].

More than a century after the parsnip webworm first appeared in North America, reconstructing its history of westward and southward expansion is challenging, but, because of its status as an economic pest of cultivated parsnip, its first appearance in an area previously free of the pest often merited collection and preservation in a museum or a report in a journal. A survey of the literature and museum records relating to the expanding geographic distribution of the parsnip webworm reveals that *D. radiella* successfully expanded its range from the East Coast to the West coast in at most 52 years, between 1862 to 1914. In 1925, *D. radiella* was reported to have been established Yuma County in Arizona, bordering Mexico, which represents considerable southward expansion of its range (Clarke, 1941) [38]. By 1932, Dustan (1932) [39] noted that this species was present in “all provinces” of Canada. Given that *P. sativa* was a frequently planted crop in the early part of the *D. radiella* colonization of North America [40], it remains to be seen whether *D. depressana* will be able to expand its range at the same pace as *D. radiella*. Currently, the range of *D. depressana* extends west to Iowa, 14 years after its initial discovery in 2008 in Ontario. If Iowa is in fact the western edge only 14 years after its introduction, *D. depressana* is already outpacing *D. radiella* in colonizing North America. Such rapid range expansion may not be entirely attributable to biological traits; many more opportunities for rapid human-assisted dispersal exist today, for example, than existed for *D. radiella* in the 19th century. Notwithstanding, by virtue of its increased period of seasonal activity and the ubiquity of available hosts, particularly *D. carota,* along North American highways, *D. depressana* may indeed require less time to colonize the continent.

This study is unique in that *D. depressana* is still apparently in the early stages of its invasion, whereas many similar studies assess population structure long after invasive species have established (but see Nayyar et al., 2021) [41]. Invasive insects with relatively broad host ranges often have profound impacts on ecosystem dynamics, not only through imposing additional selection pressure on their food plants, but also in terms of competitive displacement of native insects and introduction of new pathogens [42]. Whereas *D. radiella*, a superspecialist herbivore, reestablished its tightly coevolved relationship with *P. sativa* in North America [18,40], subsequently colonizing a single new native host *Heracleum maximum* (a congener of European host species) [20] as far as is known, *D. depressana* offers a novel comparator for assessing the ecological impact of less specialized Lepidoptera. One avenue of exploration would be to monitor the impact of *D. depressana* on wild carrot, *D. carota*, a non-native but long-established umbellifer with low levels of furanocoumarin production [43]. Intense florivory by *D. depressana* may select for greater production of furanocoumarins, as *D. radiella* did in *P. sativa* within 20 years of its introduction to North America [15], potentially altering the community structure of native insects consuming *D. carota* and may similarly affect native umbellifers, such as golden alexanders, *Zizia aurea*, the seeds of which are consumed by this species in the laboratory (Dean et al., submitted).

## 5. Conclusions

In conclusion, comparing the invasion history of two congeneric species, which share some but not all ecological traits, provides a unique opportunity to gain insights into the extent to which certain life history traits affect the likelihood of acquiring novel hostplants in the area of introduction. Key shared ecological and behavioral traits include areas of indigeneity (i.e., Eurasia), utilization of hostplants in the Apiaceae, specialized consumption of flowers and immature fruits, and use of silken webbing to aggregate umbels. In contrast, differences in ecology and behavior between the two congeners include voltinism, per-plant densities within umbels, and diet breadth. Identifying which of these differences, if any, contribute to the degree of genetic differentiation across landscapes may have predictive power in anticipating the rate at which new communities are invaded and the extent of the threat they present to the native flora.

## Figures and Tables

**Figure 1 insects-13-00789-f001:**
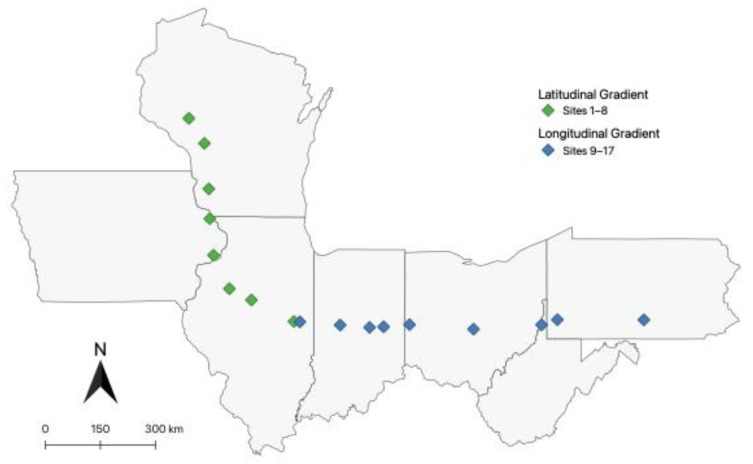
Map depicting the sampling sites of both collection trips. The latitudinal gradient (green) comprises 8 sites from Urbana, IL to Eau Claire, WI (~680 km). The longitudinal gradient (blue) comprises 9 sites from Urbana, IL to Newburg, PA (~1078 km).

**Figure 2 insects-13-00789-f002:**
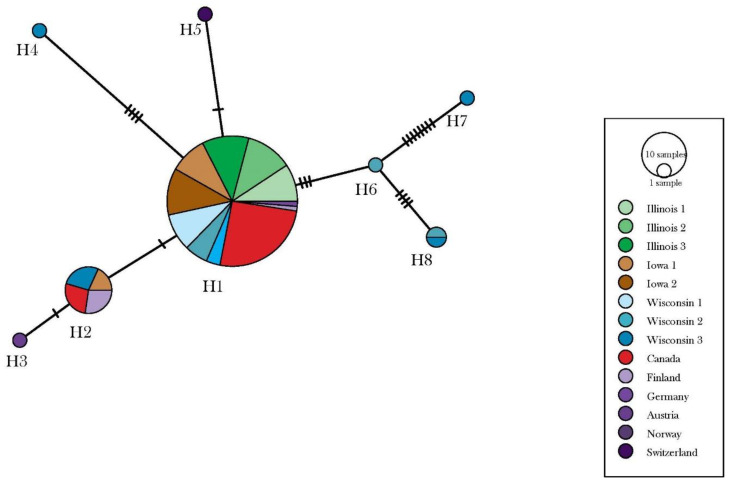
TCS (Templeton-Crandall-Sing) network depicting relationships between COI haplotypes of *Depressaria depressana* including those collected from sites sampled along a latitudinal gradient from Urbana, IL to Eau Claire, WI and retrieved from the BOLD database. The size of each circle represents number of individuals belonging to each haplotype (H1–7). The number of hash marks on branches between haplotypes represent the number of mutational steps.

**Figure 3 insects-13-00789-f003:**
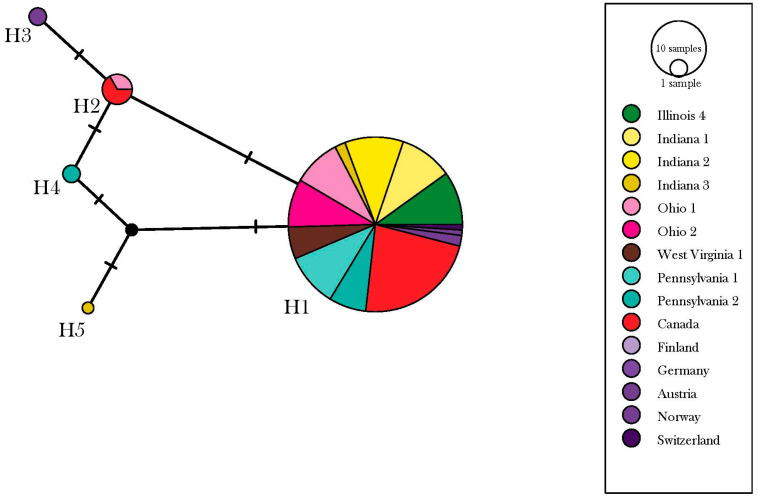
TCS (Templeton-Crandall-Sing) network depicting relationships between COI haplotypes of *Depressaria*
*depressana* including those collected from sites sampled along a longitudinal gradient from Urbana, IL to Newburg, PA and retrieved from the BOLD database. The size of each circle represents number of individuals belonging to each haplotype (H1–6). The number of hash marks on branches between haplotypes represent the number of mutational steps.

**Figure 4 insects-13-00789-f004:**
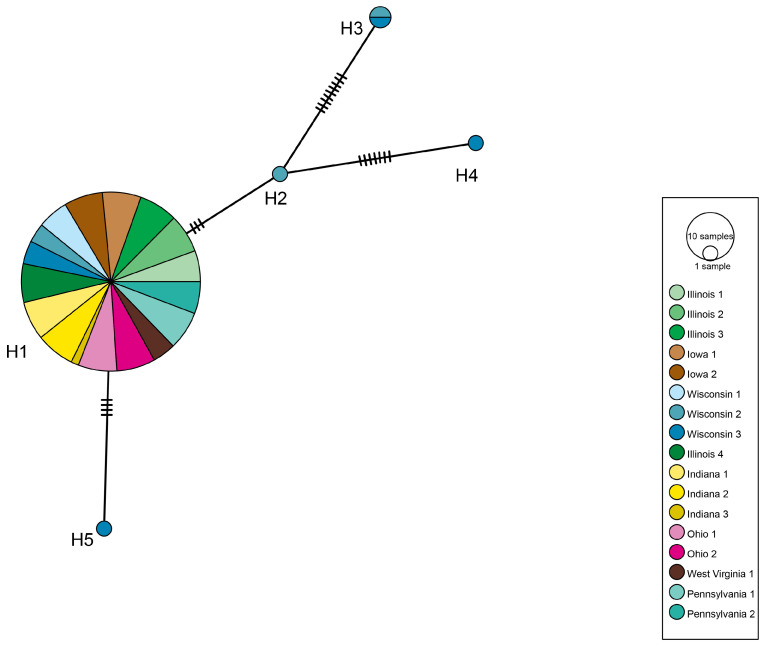
TCS (Templeton-Crandall-Sing) network depicting relationships between COI haplotypes of *Depressaria*
*depressana* collected from sites sampled along both the latitudinal gradient from Urbana, IL to Eau Claire, WI and the longitudinal gradient from Urbana, IL to Newburg, PA. The size of each circle represents number of individuals belonging to each haplotype (H1–5). The number of hash marks on branches between haplotypes represent the number of mutational steps.

**Figure 5 insects-13-00789-f005:**
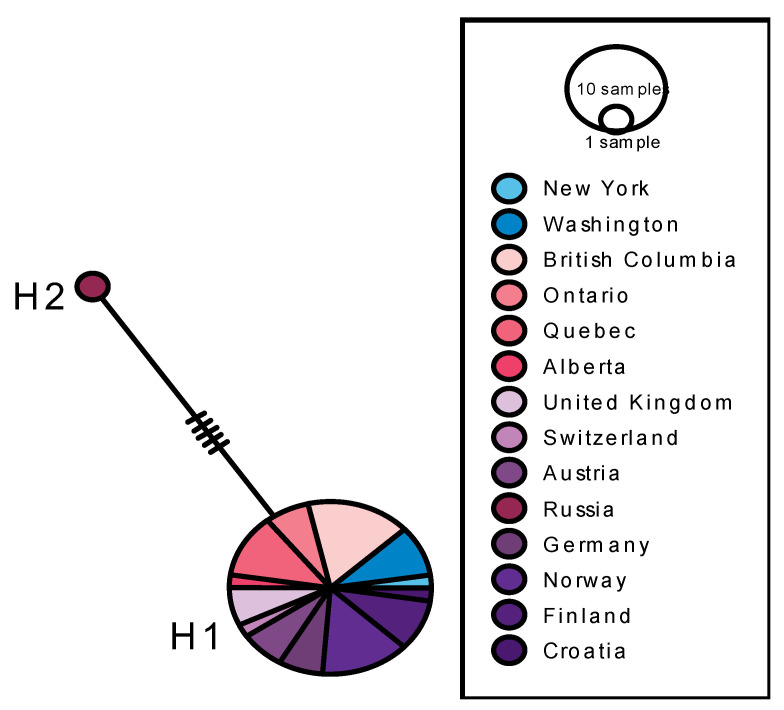
TCS (Templeton-Crandall-Sing) network depicting relationships between COI haplotypes of *Depressaria*
*radiella*. Partial COI sequences for all individuals in this analysis were retrieved from the BOLD database. The size of each circle represents number of individuals belonging to each haplotype (H1–2). The number of hash marks on branches between haplotypes represent the number of mutational steps.

**Table 1 insects-13-00789-t001:** Sampling locations and GPS information for *Depressaria depressana* collected from wild carrot (*Daucus carota*) for this study.

Site	State	Latitude	Longitude	Collection Date
1	Illinois 1	40.12854	−88.1439	July, 2020
2	Illinois 2	40.62144	−89.4234	July, 2020
3	Illinois 3	40.87883	−90.0877	July, 2020
4	Iowa 1	41.63711	−90.5679	July, 2020
5	Iowa 2	42.45517	−90.6786	July, 2020
6	Wisconsin 1	43.112	−90.7066	July, 2020
7	Wisconsin 2	44.10483	−90.8402	July, 2020
8	Wisconsin 3	44.64364	−91.3084	July, 2020
9	Illinois 4	40.11962	−87.9631	July, 2021
10	Indiana 1	40.04634	−86.7499	July, 2021
11	Indiana 2	39.99172	−85.8628	July, 2021
12	Indiana 3	40.00483	−85.4401	July, 2021
13	Ohio 1	40.05559	−84.6591	July, 2021
14	Ohio 2	39.95099	−82.7309	July, 2021
15	West Virginia 1	40.05095	−80.68	July, 2021
16	Pennsylvania 1	40.16912	−80.2041	July, 2021
17	Pennsylvania 1	40.16586	−77.5982	July, 2021

**Table 2 insects-13-00789-t002:** Sites and sequence variation in *Depressaria radiella*. Variable alignment positions are 136, 302, 337, 578, 508, 592, 628, respectively.

					Total	Variable Positions
		Country	State/Prov	BOLD ID	Bases	AGATAAT
COMPLETE SEQUENCES	EUROPE	Austria	NOE	DEEUR158-11	658	.......
		Austria	TIR	LEATJ1038-15	658	.......
		Austria	VOR	LEATC528-13	658	.......
		Finland	Ab	LEFIF849-10	658	.......
		Finland	Al	LEFID175-10	658	.......
		Germany	BY	FBLMU176-09	658	.......
		Germany	BY	FBLMU177-09	658	.......
		Germany	BY	FBLMU703-09	658	.......
		Norway	Aker	LEPVM100-12	658	.......
		Norway	Aust-Ag	LEPVM101-12	658	.......
		Norway	Vest	LON2555-15	658	.......
		Russia	Caucasus	DEEUR477-15	658	GACCG..
		Switzerland	VS	DEEUR045-11	658	.......
		UK	ENG	CGUKA083-09	658	.......
		UK	ENG	CGUKC898-09	658	.......
	N AMERICA	UK	ENG	CGUKD655-09	658	.......
		Canada	BC	LALPA1354-12	658	.......
		Canada	BC	LALPA1355-12	658	.......
		Canada	BC	LBCW061-08	658	.......
		Canada	Ont	MECB697-05	658	.......
		Canada	Que	MEC086-04	658	.......
		Canada	Que	RDLQD598-06	658	.......
		Canada	Que	RDLQD599-06	658	.......
		USA	NY	LNAUT2820-14	658	.......
		USA	WA	EHL851-12	658	.......
		USA	WA	RWWC908-12	658	.......
PARTIAL SEQUENCES	EUROPE	Croatia	Lika-Senj	LON6919-18	632	......G
		Finland	Ab	LEFIF848-10	656	.......
		Finland	Al	LEFIB073-10	647	.......
		Norway	Trond	GMNWK4200-14	618	.......
		Norway	Trond	GMNWL3227-14	567	.......
		Norway	Trond	GMNWK4187-14	534	.......
	N AMERICA	Canada	Alta	SMTPM6085-15	588	.......
		Canada	BC	LOWCE394-06	617	.......
		Canada	BC	LOWCE442-06	612	.......
		Canada	BC	LOWCE437-06	610	.......
		Canada	BC	LOWCE397-06	599	.......
		Canada	Ont	SMTPB5733-13	543	.......
		Canada	Ont	MEC171-04	642	.....GG
		Canada	Que	MEC339-04	629	.......
		Canada	Que	RDLQD600-06	649	.......
		USA	NY	LNAUT2821-14	307	.......
		USA	WA	RWWC1236-13	591	.......
		USA	WA	RWWC1336-14	585	.......
		USA	WA	RWWC1331-14	579	.......

**Table 3 insects-13-00789-t003:** Reconstruction of the establishment of *D. radiella* across the USA and Canada with the use of literature and museum records.

State/Province	Date	Type	Record
Ontario	1862	Personal Observation	Bethune (1869)
Pennsylvania	1888	Collection Record	Collected by T Pergande, reported in Clarke (1941)
“Eastern United States”	1889	Personal Observation	Riley (1889)
Michigan	1890	Preserved Specimen	The Albert J. Cook Arthropod Research Collection
Illinois	1900	Collection Record	Collected by WD Kearfott, reported in Clarke (1941)
Quebec	1903	Collection Record	Collected by CH Young, reported in Clarke (1941)
Utah	1907	Collection Record	Collected by ESG Titus, reported in Clarke (1941)
Oregon	1914	Collection Record	Collected by L Leland, reported in Clarke (1941)
Nova Scotia	1915	Collection Record	No Collector Credited, reported in Clarke (1941)
Connecticut	1919	Preserved Specimen	The Yale Peabody Museum
Massachusetts	1920	Collection Record	Collected by JD Caffrey, reported in Clarke (1941)
Rhode Island	1920	Collection Record	No Collector Credited, reported in Clarke (1941)
Maine	1923	Preserved Specimen	The Yale Peabody Museum
Arizona	1925	Collection Record	Collected by OC Poling, reported in Clarke (1941)
British Columbia	1925	Collection Record	Collected by LE Marmont, reported in Clarke (1941)
Washington	1930	Collection Record	Collected by WW Baker, reported in Clarke (1941)
Indiana	1931	Collection Record	Collected by GS Walley, reported in Clarke (1941)
“All provinces of Canada”	1932	Personal Observation	Dustan (1932)
New York	1939	Collection Record	Collected by JFG Clarke, reported in Clarke (1941)
Minnesota	1947	Preserved Specimen	The University of Minnesota Insect Collection
Kentucky	1955	Collection Record	Evidence of webworm activity in herbarium sample, reported in Zangerl and Berenbaum (2005)
Ohio	1961	Preserved Specimen	The Cleveland Museum of Natural History Invertebrate Zoology Collection

## Data Availability

DNA sequence data analyzed in this study are publicly available in BOLD database and in Genbank.

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
