# Peer review of "Genetic Structure and Colonization of North America by Depressaria depressana (Fabricius 1775) (Lepidoptera: Depressariidae) over 15 Years; Contrasts with Westward Expansion of Depressaria radiella (Goeze, 1783) over 160 Years"

_insects, 2022, doi:10.3390/insects13090789_

Round 1
Reviewer 1 Report
This study is devoted to a genetic structure and diversity of two invasive species, members of Depressariidae family, namely Depressaria depressana and Depressaria radiella. The MS is well-written and undoubtedly provide new and interesting data and should be accepted for publication after minor revision. Please see comments below, and I hope that these may be useful.
General comment
I would recommend authors to add a map showing distribution of species in question (D. depressana and D. radiella) in Palaearctic and North America, or at least discuss the distribution range in the Introduction section.
Minor comments
Title
I suggest adding authors of the descriptions and a higher classification to Latin names:
Genetic structure and colonization of North America by Depressaria depressana (Fabricius 1775) (Lepidoptera: Depressariidae) over 15 years; contrasts with westward expansion of Depressaria radiella (Goeze, 1783) over 160 years
Lines 14-16
The purple carrot-seed moth Depressaria depressana, a European species first reported in North America in 2008, specializes on reproductive structures of species throughout the family Apiaceae and thus presents a potential threat to native North American umbellifers.
In order to avoid same words I would recommend to change species either to “taxon” in the first case (a European taxon first reported in North America in 2008), or to “taxa” in the second case (specializes on reproductive structures of taxa).
Lines 34-35
“…by constructing haplotype networks using COI markers…” should be singular, COI marker since the only one marker (DNA barcode) was used in the analysis
Line 60
“…and adaptation to, new environments…” remove comma.
Line 72
Please, add authors of the description while mention taxa in the text for the first time.
(formerly Depressaria pastinacella Duponchel, 1838)
Line 73
(Bethune 1869, named D.ontariella and subsequently synonymized with the European D. heracliana).
Not Bethune 1870? Please check. Please add Linnaeus, 1758 after D. heracliana.
Line 76
Heracleum and Pastinaca… Please, add authors.
Line 78
Pastinaca sativa… Please add authors
Lines 165-167
The partial sequence of COI was amplified with the following primers from Kim et al. (2016): (LCO) 5’-GGTCAACAAATCATAAAGATATTGG 3’ and (HCO) 5’ TAAACTTCAGGGTGACCAAAAAATCA 3’ generating a 658 bp amplicon.
It is better to provide original reference. Primers LCO1490 and HCO2198 were suggested for the first time as universal primers for invertebrates by Folmer et al. in 1994.
Lines 190-191
Genetic structuring across sampling sites was assessed across geographic space via analysis of molecular variance (AMOVA) with PopART (citation? Version?)
Please, provide citation and version as stated in your draft.
Line 192
A Mantel test was also performed using zt (version?)
Please, provide version as stated in your draft.
Lines 210-211
Across the ~680 km latitudinal gradient ranging from Urbana, IL, to Eau Claire, WI, 79 COI sequences were generated from D. depressana collected across eight geographic locations.
“…were generated from D. depressana specimens (or samples)…” please add word specimens, or samples after latin name
Lines 212-213
COI sequences from the 34 additional taxa retrieved from the BOLD database were incorporated into the 681-bp nucleotide alignment.
Primers pair HCO+LCO amplify 658-bp fragment. “681-bp nucleotide alignment” means it contains primers sequences, which has to be removed (trimmed) from the data set.
Lines 214-215
Within this alignment, 22 polymorphisms were identified as informative for the delineation of haplotypes
I suggest using term “polymorphic sites” instead of polymorphisms
Lines 217-218
Low levels of genetic diversity were detected across all taxa and the AMOVA revealed little genetic structuring (ɸST = 0.095, p = 0.059).
Level instead of levels.
In this paragraph you discussing the only one taxon, D. depressana. Thus, term taxa is impropriate here. Moreover, it`s about samples, not taxa or taxon.
Lines 219-220
Analysis of the ~1078 km longitudinal gradient ranging from Urbana, IL, to New-219 burg, PA, incorporated 88 D. depressana collected from nine geographic locations.
“…D. depressana specimens (or samples)…”
Lines 222-223
Seven informative polymorphisms were identified, 222 and taxa were clustered into six distinct haplotypes.
I suggest using term “polymorphic sites” instead of polymorphisms
Line 223.
low levels of genetic structure was observed
“low levels… were observed”, or “low level… was observed” (the second variant is preferable)
Lines 225-226
Combining the D. depressana COI datasets obtained from samples collected along the latitudinal and longitudinal gradients revealed 23 informative polymorphisms
I suggest using term “polymorphic sites” instead of polymorphisms
Lines 228
also revealed low levels of genetic structure
low level
Lines 230-231
Isolation by distance is has little to no effect of genetic variation
Isolation by distance has little to no effect of genetic variation.
Lines 235-236
Twenty-six complete COI sequences were found, 16 from Europe and 10 from North America.
Term “complete COI sequences” somewhat misleading. I guess you mean complete DNA barcodes (658 bp), rather than complete COI (>1500 bp)
Lines 237-238
The 26 complete sequences were aligned with NCBI Multiple Se-237 quence Alignment Viewr 1.22.0.
ViewEr
Lines 238-239
Is is clear from Table 2, the resulting haplotype networks revealed two distinct haplotypes, with 5 informative SNPs distinguishing them.
There is only one network of D. radiella
The resulting haplotype networks revealed two distinct haplotypes, with 5 informative SNPs distinguishing them (see Table 2 and figure 5).
Lines 239-240
All individuals fell into the H1 haplotype, with the single D. radiella collected in Russia constituting the H2 haplotype.
All individuals fell into the H1 haplotype, with the single D. radiella specimen collected in Russia constituting the H2 haplotype.
Table 2
Russia Kaukasus
Russia, Caucasus
Figures 1,2,3,4,5
1) Legend should be in the same style, e.g. use dots instead of colons (Figure 3. Figure 3. Figure 5. NOT Figure 3: Figure 4: Figure 5: )
2) Use full latin names (Depressaria depressana) as in Figure 2, or shortened latin names (D. depressana) as in Figures 3, 4. 5
Line 284
Spodoptera frugiperda
Please, add authors of the description
Author Response
Below is the text of the review from reviewer 1; our responses appear in italicized text.
Open review; Comments and Suggestions for Authors
This study is devoted to a genetic structure and diversity of two invasive species, members of Depressariidae family, namely Depressaria depressana and Depressaria radiella. The MS is well-written and undoubtedly provide new and interesting data and should be accepted for publication after minor revision. Please see comments below, and I hope that these may be useful.
General commentI would recommend authors to add a map showing distribution of species in question (D. depressana and D. radiella) in Palaearctic and North America, or at least discuss the distribution range in the Introduction section.
We thank the reviewer for the kind words and the recommendation for acceptance after minor revision. In terms of the suggestion to add a map showing the distribution range, we feel that information already in the Introduction and Discussion and the images in the Graphical Abstract convey a sufficient sense of the distributional range of these two species.
Minor comments
Title I suggest adding authors of the descriptions and a higher classification to Latin names: Genetic structure and colonization of North America by Depressaria depressana (Fabricius 1775) (Lepidoptera: Depressariidae) over 15 years; contrasts with westward expansion of Depressaria radiella (Goeze, 1783) over 160 years
The authors agree with this suggestion. The change has been incorporated into the manuscript.
Lines 14-16 The purple carrot-seed moth Depressaria depressana, a European species first reported in North America in 2008, specializes on reproductive structures of species throughout the family Apiaceae and thus presents a potential threat to native North American umbellifers.In order to avoid same words I would recommend to change species either to “taxon” in the first case (a European taxon first reported in North America in 2008), or to “taxa” in the second case (specializes on reproductive structures of taxa).
The authors agree with this suggestion. The change has been incorporated into the manuscript.
Lines 34-35…by constructing haplotype networks using COI markers…” should be singular, COI marker since the only one marker (DNA barcode) was used in the analysis
The authors agree with this suggestion. The change has been incorporated into the manuscript.
Line 60“…and adaptation to, new environments…” remove comma.
The authors agree with this suggestion. The change has been incorporated into the manuscript.
Line 72 Please, add authors of the description while mention taxa in the text for the first time.(formerly Depressaria pastinacella Duponchel, 1838)
The authors agree with this suggestion. The change has been incorporated into the manuscript.
Line 73 (Bethune 1869, named D.ontariella and subsequently synonymized with the European D. heracliana). Not Bethune 1870? Please check. Please add Linnaeus, 1758 after D. heracliana.
Confirmed that is was Bethune 1869. The second suggestion has been incorporated into the manuscript.
Line 76 Heracleum and Pastinaca… Please, add authors.
The authors agree with this suggestion. The changes have been incorporated into the manuscript.
Line 78 Pastinaca sativa… Please add authors
The authors agree with this suggestion. The change has been incorporated into the manuscript.
Lines 165-167 The partial sequence of COI was amplified with the following primers from Kim et al. (2016): (LCO) 5’-GGTCAACAAATCATAAAGATATTGG 3’ and (HCO) 5’ TAAACTTCAGGGTGACCAAAAAATCA 3’ generating a 658 bp amplicon. It is better to provide original reference. Primers LCO1490 and HCO2198 were suggested for the first time as universal primers for invertebrates by Folmer et al. in 1994.
The authors agree with this suggestion. The original author was cited.
Lines 190-191 Genetic structuring across sampling sites was assessed across geographic space via analysis of molecular variance (AMOVA) with PopART (citation? Version?) Please, provide citation and version as stated in your draft.
The authors agree with this suggestion. The changes have been incorporated into the manuscript.
Line 192 A Mantel test was also performed using zt (version?) Please, provide version as stated in your draft.
The authors agree with this suggestion. The change has been incorporated into the manuscript.
Lines 210-211 Across the ~680 km latitudinal gradient ranging from Urbana, IL, to Eau Claire, WI, 79 COI sequences were generated from D. depressana collected across eight geographic locations.“…were generated from D. depressana specimens (or samples)…” please add word specimens, or samples after latin name
The authors agree with this suggestion. The change has been incorporated into the manuscript.
Lines 212-213 COI sequences from the 34 additional taxa retrieved from the BOLD database were incorporated into the 681-bp nucleotide alignment. Primers pair HCO+LCO amplify 658-bp fragment. “681-bp nucleotide alignment” means it contains primers sequences, which has to be removed (trimmed) from the data set.
The primer regions have been trimmed from the dataset and the number of bps has been adjusted accordingly in the manuscript.
Lines 214-215 Within this alignment, 22 polymorphisms were identified as informative for the delineation of haplotypes. I suggest using term “polymorphic sites” instead of polymorphisms
The authors agree with this suggestion. The change has been incorporated into the manuscript.
Lines 217-218 Low levels of genetic diversity were detected across all taxa and the AMOVA revealed little genetic structuring (ɸST = 0.095, p = 0.059). Level instead of levels. In this paragraph you discussing the only one taxon, D. depressana. Thus, term taxa is impropriate here. Moreover, it`s about samples, not taxa or taxon.
The authors agree with this suggestion. The change has been incorporated into the manuscript.
Lines 219-220 Analysis of the ~1078 km longitudinal gradient ranging from Urbana, IL, to New-219 burg, PA, incorporated 88 D. depressana collected from nine geographic locations. “…D. depressana specimens (or samples)…”
The authors agree with this suggestion. The change has been incorporated into the manuscript.
Lines 222-223 Seven informative polymorphisms were identified, 222 and taxa were clustered into six distinct haplotypes. I suggest using term “polymorphic sites” instead of polymorphisms
The authors agree with this suggestion. The change has been incorporated into the manuscript.
Line 223. low levels of genetic structure was observed “low levels… were observed”, or “low level… was observed” (the second variant is preferable)
The authors agree with this suggestion. The change has been incorporated into the manuscript.
Lines 225-226 Combining the D. depressana COI datasets obtained from samples collected along the latitudinal and longitudinal gradients revealed 23 informative polymorphisms I suggest using term “polymorphic sites” instead of polymorphisms
The authors agree with this suggestion. The change has been incorporated into the manuscript.
Lines 228 also revealed low levels of genetic structure--low level
The authors agree with this suggestion. The change has been incorporated into the manuscript.
Lines 230-231
Isolation by distance is has little to no effect of genetic variation. Change to Isolation by distance has little to no effect of genetic variation.
The authors agree with this suggestion. The change has been incorporated into the manuscript.
Lines 235-236 Twenty-six complete COI sequences were found, 16 from Europe and 10 from North America. Term “complete COI sequences” somewhat misleading. I guess you mean complete DNA barcodes (658 bp), rather than complete COI (>1500 bp)
The authors agree with this suggestion. The wording has been adjusted accordingly.
Lines 237-238 The 26 complete sequences were aligned with NCBI Multiple Se-237 quence Alignment Viewr 1.22.0. ViewEr
The authors have incorporated this correction into the manuscript.
Lines 238-239 Is is clear from Table 2, the resulting haplotype networks revealed two distinct haplotypes, with 5 informative SNPs distinguishing them.There is only one network of D. radiella The resulting haplotype networks revealed two distinct haplotypes, with 5 informative SNPs distinguishing them (see Table 2 and figure 5).
The authors agree with this suggestion. The changes have been incorporated into the manuscript.
Lines 239-240 All individuals fell into the H1 haplotype, with the single D. radiella collected in Russia constituting the H2 haplotype. Change to: All individuals fell into the H1 haplotype, with the single D. radiella specimen collected in Russia constituting the H2 haplotype.
The authors agree with this suggestion. The change has been incorporated into the manuscript.
Table 2
Russia Kaukasus
Russia, Caucasus
The authors agree with this suggestion. The change has been incorporated into the manuscript.
Figures 1,2,3,4,5
1) Legend should be in the same style, e.g. use dots instead of colons (Figure 3. Figure 3. Figure 5. NOT Figure 3: Figure 4: Figure 5: )
The authors agree with this suggestion. The change has been incorporated into the manuscript.
2) Use full latin names (Depressaria depressana) as in Figure 2, or shortened latin names (D. depressana) as in Figures 3, 4. 5
The authors agree with this suggestion. The change has been incorporated into the manuscript.
Line 284
Spodoptera frugiperda Please, add authors of the description
The authors agree with this suggestion. The change has been incorporated into the manuscript.
Reviewer 2 Report
This is an interesting manuscript, in particular due to a cleverly posed
problem for the investigation. Of course, the topic of pests and their
invasion is of a practical importance, but I was more impressed with the
system of two closely related species that invaded the US 150 years apart.
Comparative study of their genetic variation promises to answer many
interesting and general questions about the species evolution in new
environments and the relationship between polyphagy and genetic diversity.
The study is described in detail and the methods are clear. It is a
well-written manuscript. The major observation is that the COI haplotype
diversity of a polyphagous species is much larger than that of a
specialist and this higher diversity/polyphagy is correlated with faster
colonization. This is an interesting conclusion that deserves to be
published. One question is that usually higher genetic diversity is
correlated with larger population size. If known or estimated, how do the
numbers of individuals of these species compare in various populations? Is
it true that Depressaria depressana has larger population sizes in
general?
Unfortunately, variation in the COI barcode fragment is not particularly
extensive, which limits the power of this study. Therefore, the title may
promise more than what the study is about. Maybe it would make sense to
replace the word "genetic" with the word "haplotype"? I think it will be
really interesting to study this system with more extensive, genomic-scale
sequencing, which hopefully the authors will pursue in their future
projects.
A minor comment: While Figs. 2-4 show round circles, they appear
compressed in Fig. 5. For consistency, it could be nice to avoid this
vertical compression.
Author Response
Reviewer comments appear below in plain text and actions taken in response in italicized text.
Comments and Suggestions for Authors
This is an interesting manuscript, in particular due to a cleverly posed problem for the investigation. Of course, the topic of pests and their invasion is of a practical importance, but I was more impressed with the system of two closely related species that invaded the US 150 years apart. Comparative study of their genetic variation promises to answer many interesting and general questions about the species evolution in new environments and the relationship between polyphagy and genetic diversity. The study is described in detail and the methods are clear. It is a well-written manuscript. The major observation is that the COI haplotype diversity of a polyphagous species is much larger than that of a specialist and this higher diversity/polyphagy is correlated with faster colonization. This is an interesting conclusion that deserves to be published. One question is that usually higher genetic diversity is correlated with larger population size. If known or estimated, how do the
numbers of individuals of these species compare in various populations? Is it true that Depressaria depressana has larger population sizes in general? Unfortunately, variation in the COI barcode fragment is not particularly
extensive, which limits the power of this study. Therefore, the title may promise more than what the study is about. Maybe it would make sense to replace the word "genetic" with the word "haplotype"? I think it will be
really interesting to study this system with more extensive, genomic-scale sequencing, which hopefully the authors will pursue in their future projects.
We thank this reviewer for the kind words about our work and we hope to continue this work in exactly the way the reviewer recommends! With respect to the title, the authors believe that, because haplotype variation at the CO1 locus constitutes genetic structure, the wording of the title is still accurate and we hope to continue to use it as originally written.
A minor comment: While Figs. 2-4 show round circles, they appear compressed in Fig. 5. For consistency, it could be nice to avoid this vertical compression.
The authors agree with this suggestion. Figure 5 has been updated to remove vertical compression.